# Cardiac Dose Predicts the Response to Concurrent Chemoradiotherapy in Esophageal Squamous Cell Carcinoma

**DOI:** 10.3390/cancers15184580

**Published:** 2023-09-15

**Authors:** Yu-Chieh Ho, Yuan-Chun Lai, Hsuan-Yu Lin, Ming-Hui Ko, Sheng-Hung Wang, Shan-Jun Yang, Tsai-Wei Chou, Li-Chung Hung, Chia-Chun Huang, Tung-Hao Chang, Jhen-Bin Lin, Jin-Ching Lin

**Affiliations:** 1Department of Radiation Oncology, Changhua Christian Hospital, Changhua 500, Taiwan; 181009@cch.org.tw (Y.-C.H.); 169336@cch.org.tw (S.-J.Y.); 125587@cch.org.tw (T.-W.C.); 117784@cch.org.tw (L.-C.H.); 101878@cch.org.tw (C.-C.H.); 81006@cch.org.tw (T.-H.C.); 182715@cch.org.tw (J.-C.L.); 2Division of Medical Physics, Department of Radiation Oncology, Changhua Christian Hospital, Changhua 500, Taiwan; 103424@cch.org.tw (Y.-C.L.); 78990@cch.org.tw (M.-H.K.); 3Department of Medical Imaging Radiological Science, Central Taiwan University of Science and Technology, Taichung 406, Taiwan; 4Division of Haematology/Oncology, Department of Internal Medicine, Changhua Christian Hospital, Changhua 500, Taiwan; 93687@cch.org.tw; 5Department of Radiation Oncology, Lukang Christian Hospital, Changhua Christian Medical Foundation, Lukang 505, Taiwan; 719260@cch.org.tw; 6Department of Medical Imaging and Radiological Technology, Yuanpei University of Science and Technology, Hsinchu 300, Taiwan; 7Research Department, Division of Translation Research, Changhua Christian Hospital, Changhua 500, Taiwan; 8Institute of Clinical Medicine, School of Medicine, National Yang-Ming Chiao-Tung University, Taipei 112, Taiwan

**Keywords:** esophageal cancer, concurrent chemoradiation, radiation dosimetry, survival outcomes

## Abstract

**Simple Summary:**

Pursuing a maximal clinical response for esophageal cancer after definitive chemoradiation is crucial, as it may be an integral surrogate endpoint for survival. In addition, radiation dosimetry parameters and treatment-induced inflammation have been validated in terms of survival outcomes. This study evaluated the treatment response and clinical variables to determine whether there were correlations between them. In non-surgical candidates, the optimization of radiation techniques to spare cardiac irradiation should be emphasized.

**Abstract:**

Definitive concurrent chemoradiation (CCRT) is the standard treatment for cervical esophageal cancer and non-surgical candidates. Initial treatment response affects survival; however, few validated markers are available for prediction. This study evaluated the clinical variables and chemoradiation parameters associated with treatment response. Between May 2010 and April 2016, 86 completed CCRT patients’ clinical, dosimetric, and laboratory data at baseline and during treatment were collected. Cox regression analysis assessed the risk factors for overall survival (OS). A receiver operating characteristic curve with Youden’s index was chosen to obtain the optimal cut-off value of each parameter. Treatment response was defined per Response Evaluation Criteria in Solid Tumors v.1.1 at the first post-CCRT computed tomography scan. Responders had complete and partial responses; non-responders had stable and progressive diseases. Logistic regression (LR) was used to evaluate the variables associated with responders. The Cox regression model confirmed the presence of responders (*n* = 50) vs. non-responders (*n* = 36) with a significant difference in OS. In multivariate LR, cardiac dose–volume received ≥10 Gy; the baseline hemoglobin level, highest neutrophil to lymphocyte ratio during CCRT, and cumulative cisplatin dose were significantly associated with the responders. The initial clinical treatment response significantly determines disease outcome. Cardiac irradiation may affect the treatment response.

## 1. Introduction

Esophageal cancer is a major worldwide threat with a poor prognosis. It is unevenly distributed, with the highest incidence and mortality rates in Eastern Asia and Southern and Eastern Africa. Although the incidence of adenocarcinoma is increasing in developed countries, squamous cell carcinoma remains the predominant subtype [1]. In Taiwan, its incidence has increased substantially during the past few years, predominantly in males [2].

Since the seminal Chemoradiotherapy for Oesophageal Cancer Followed by Surgery Study (CROSS), preoperative concurrent chemoradiation (CCRT) has been recommended as a standard treatment for esophageal cancer [3]. Adjuvant immunotherapy can further improve survival outcomes in patients with residual disease after preoperative treatment, following clear surgical resection [4]. However, not every patient is fit or willing to undergo radical, life-changing organ resection. As the pathological response rate is high after CCRT in patients with esophageal squamous cell carcinoma [3], definitive CCRT with salvage esophagectomy in selective cases is also considered a validated treatment with curative intent [5,6,7,8]. There are ongoing prospective trials to validate the concept of organ preservation [9,10].

For patients who receive definitive CCRT as non-operative management, the intermediate treatment endpoint, the clinical treatment response, is crucial for survival and may guide subsequent treatment decisions [11,12]. However, there are limited data on which clinical factors predict a clinical response. Radiation dosimetry factors such as cardiac and lung doses and peri-treatment hematological changes have drawn attention through their correlation with survival [13,14,15,16,17]. The total cardiac dose, and that administered to cardiac substructures, is a recent research hotspot that may predict major coronary events and survival [13,15,18,19,20]. This study aimed to confirm that the initial treatment response strongly drives overall survival, and to determine the clinical factors that most affect treatment response.

## 2. Materials and Methods

### 2.1. Patient Characteristics and Study Design

A retrospective review of 86 patients with non-metastatic thoracic esophageal cancer treated with non-surgical treatments, including definitive radiotherapy (RT) with or without induction, and concurrent or adjuvant chemotherapy, was performed at our institution. All patients completed treatment between May 2010 and April 2016, including a complete course of RT and at least one follow-up evaluation with chest computed tomography (CT).

All patients had esophagogastroduodenoscopy (EGD) biopsy-proven squamous cell carcinoma, and the gross tumor extent was calculated using a scope. The 7th edition of the American Joint Committee on Cancer TNM classification system was used, with a chest CT scan for staging. Bronchoscopic evaluation was suggested if there was any suspicious invasion of the trachea on chest CT. Whole-body F-18 fluorodeoxyglucose positron emission tomography (PET)/CT, Tc99m methylene diphosphonate bone scans, abdominal sonography, and brain magnetic resonance imaging were also options for the initial and further follow-up workup as metastatic surveys based on the physician’s discretion. Patients with any other history of cancer or synchronous cancer were excluded. We also used the age-adjusted Charlson Comorbidity Index (ACCI) score [21,22] (current esophageal cancer diagnosis was not included) to estimate the 10-year pre-treatment risk of mortality. Complete blood counts (CBC) before and during RT were analyzed. We used the Common Terminology Criteria for Adverse Events (CTCAE) version 4.0, for treatment-related toxicity grading.

The median total radiation dose was 59.4 Gy (standard deviation: 6.1 Gy) with standard daily fractionation (1.8–2.0 Gy per fraction). The concurrent chemotherapy regimen was cisplatin (75 mg/m^2^) on the first day of weeks 1 and 5 and fluorouracil (750 mg/m^2^ per day) by continuous infusion on the first 4 days of weeks 1 and 5. The chemotherapy sequence and RT were determined based on the physician’s discretion and the patient’s general condition. After treatment, chest CT and EGD were used to evaluate local, regional, and distant failure every 3–6 months, in combination with any other metastatic disease assessment, if indicated. The initial treatment response was assessed 5–12 weeks after RT, mainly based on the first chest CT follow-up image, and divided into complete response (CR), partial response (PR), stable disease, and progressive disease (PD) according to the Response Evaluation Criteria in Solid Tumors (RECIST) v1.1 criteria. If EGD revealed suspicious lesions, a biopsy confirmation was indicated. Patient follow-up was updated and censored on 31 March 2023. The Institutional Review Board of Changhua Christian Hospital approved this study as a prognostic factor analysis for esophageal cancer treated with radiotherapy (CCH IRB No.: 180310). They agreed to the waiver of informed consent due to the retrospective analysis of patient data.

### 2.2. Radiation Treatment and Dosimetric Analysis

RT covers the following as mandatory: the primary tumor and lymphadenopathy plus a 1 cm circumferential margin and a 3–5-cm longitudinal margin. The decision to cover the elective nodal region was made at the discretion of the physician. We reviewed the whole target volume and critical organs at risk (OAR), such as the heart and lungs, and recontoured (if necessary) without adding a margin to each organ. Dosimetric analyses were then performed by a dosimetrist and reviewed by a physician using RT plans available on the Pinnacle Treatment Planning System (Pinnacle Treatment Planning/Philips Radiation Oncology Systems, Fitchburg, WI, USA). The mean dose and relevant dose–volume histogram (DVH) parameters were evaluated, as the heart and lung received the relative percentages of volume for at least x (Gy), identified as Vx (%).

### 2.3. Hematological Parameters

Baseline hemoglobin (Hb), white blood cell (WBC) count, absolute neutrophil count (ANC), and absolute lymphocyte count (ALC) were collected before treatment. The neutrophil-to-lymphocyte ratio (NLR) was calculated by dividing ANC by ALC. We recorded the ALC nadir at the lowest ALC during CCRT. The highest NLR during CCRT (NLR-h) was determined on the date of CBC, using the lowest lymphocyte percentage differentiation count during RT.

### 2.4. Statistical Analysis

Continuous data are presented as median and standard deviation (SD) and categorical data as numbers and percentages. We focused on the most relevant clinical endpoint, overall survival (OS), defined as the date of diagnosis to any cause of death. Patients who did not die were censored at the last follow-up date. We used the Cox regression model to analyze the relationship between the patients and treatment characteristics and OS. The OS rate was then estimated using Kaplan–Meier analyses and the log-rank test to calculate the significance of differences in survival estimates. To test the possible factors associated with the clinical response and hematological and treatment variables, we used the receiver operating characteristic curve (ROC) with Youden’s index to obtain the optimal cut-off value of each parameter. A logistic regression model was then used to validate the variables and their relationship with clinical response. A *p*-value < 0.1 was considered acceptable in univariate analysis to further multivariate validation. A *p*-value < 0.05 was then considered statistically significant in multivariate analysis. Hazard ratios (HR) and odds ratios (OR) are reported with a 95% confidence interval (CI). All analyses were performed using SPSS version 26 software (IBM Corporation, Armonk, NY, USA).

## 3. Results

### 3.1. Patient and Treatment Characteristics

Table 1 summarizes the patient and treatment characteristics of the 86 patients enrolled between May 2010 and April 2016. All patients had squamous cell carcinoma, with the majority having a middle (37.2%) or lower (34.9%) location, grade 2 histology (69.8%), and stage III disease (70.9%). We used various radiation techniques, with either three-dimensional conformal radiation therapy (3DCRT) (23.2%), intensity-modulated radiation therapy (IMRT) (38.3%), or volumetric-modulated arc therapy (VMAT) (38.3%). The median prescription dose was 5940 cGy (SD: 605.3). The medians of mean heart and lung doses were 2047 cGy (SD: 692.5) and 1511.5 cGy (SD: 295.9), respectively.

Nearly all patients (97.7%) received concurrent chemoradiation (CCRT), and only two received chemotherapy and RT sequentially. More than half (55.8%) of patients underwent some form of induction chemotherapy before CCRT. The standard chemotherapy regimens included triweekly cisplatin and fluorouracil (96.5%). Based on the physician’s discretion and patient tolerance, the median cumulative cisplatin dose during CCRT was 75 mg/m^2^ (SD: 41.6) and the median cumulative cisplatin dose before CCRT completion was 140 mg/m^2^ (SD: 54.7). During CCRT, the most common acute toxicity of grade ≥ 3 was hematological toxicity, and a detailed analysis is shown below. The other non-hematological grade 3 toxicities observed were dysphagia (7.0%), mucositis (2.3%), fatigue (1.2%), and anorexia (1.2%); no grade ≥ 4 toxicities were recorded.

With a median follow-up of 15 months (SD: 31.9) in all patients and 96.5 months (SD: 17.3) in survivors, the estimated median OS was 15 months in all patients (95% CI: 12.729–17.271). The estimated 2- and 5-year OS values were 30.2 and 19.8%, respectively. At the last follow-up, 76 patients (88.4%) had died. Most patients (88.2%) died because of disease progression or disease-related complications like pneumonia. Others died owing to second primary cancer (6.6%), acute exacerbations of chronic obstructive pulmonary disease (2.6%), cerebral infarction (1.3%), or unknown etiology (1.3%). Specifically, 10/76 (13.2%) deceased patients noted cardiac morbidities, including myocardial infarction, cardiac dysrhythmia, ventricular dysfunction, or heart failure. Although eight patients (10.5%) were newly diagnosed after CCRT, only three (3.9%) had major cardiac events (one myocardial infarction and two heart failures), while others reported no symptomatic exacerbation until mortality. There was also no newly diagnosed cardiac disease among the survivors.

### 3.2. Hematological Parameters and Toxicities

Table 1 shows the hematological parameters. Data on pre-treatment baseline complete blood count (CBC) with differentials were available for 81 patients, with a median of 18 days between baseline CBC and RT (SD: 14.8). Median baseline Hb, WBC, ANC, ALC, and NLR were 12.8 g/dL (SD: 1.9), 7900 cells/mm^3^ (SD: 2669.1), 5074 cells/mm^3^ (SD: 2492.1), 1635 cells/mm^3^ (SD: 599.8), and 3.32 (SD: 2.2), respectively. Additionally, 9/81 (11.1%) patients had a low pre-treatment ALC level, defined as grade 1 lymphopenia in five patients and grade 2 lymphopenia in four patients.

Data on the ALC nadir during CCRT were available for all patients; we could then calculate the NLR-h. We found that the ALC nadir occurred at a median of 28 days (SD: 9.6) after the start of RT, whereas the NLR-h developed at a median of 30 days (SD: 10.8). The median ALC nadir and NLR-h were 219.4 cells/mm^3^ (SD: 172.3) and 17.1 (SD: 28.3), respectively. In 23 patients, the ALC nadir and NLR-h sampling days differed. During treatment, 23/86 (26.7%) patients were noted with grade 2, 17 (19.8%) with grade 3, and 8 (9.3%) with grade 4 leukopenia. In addition, 9 (10.5%) patients had grade 2, 39 (45.3%) had grade 3, and 38 (44.2%) had grade 4 lymphopenia at the nadir.

### 3.3. A Poor Initial Response Was Associated with Lower OS

After clinical evaluation of the initial treatment response, 4 (4.7%) patients achieved CR, 46 (53.5%) patients had PR, 14 (16.3%) patients had stable disease, and 22 (25.6%) had PD. Table 1 summarizes the results of univariate analysis of the factors associated with OS.

Sex, history of smoking and alcohol use, performance status, NLR-h, heart V10, and treatment response were significantly related to OS in the COX regression model. In the multivariate analysis presented in Table 1, male gender (HR 9.105, 95% CI 2.379–34.853, *p* = 0.001), poorer performance status (HR 3.755, 95% CI 1.477–9.547, *p* = 0.005), higher NLR-h (HR 1.012, 95% CI 1.002–1.022, *p* = 0.016), and non-responders (HR 4.172, 95% CI 2.296–7.583, *p* < 0.001) remained related to OS.

In addition, we used Kaplan–Meier analysis to evaluate the survival of responders (*n* = 50) vs. non-responders (*n* = 36). The median OS rates stratified by responders and non-responders were 22 (95% CI, 16.803–27.197) and 10 (95% CI, 7.060–12.940) months, respectively (*p* < 0.001, Figure 1). 

The estimated two- and five-year OS rates were 46 vs. 8.3% and 32 vs. 2.8% when stratified by responders vs. non-responders, respectively.

### 3.4. Patient Characteristics and Dosimetric Parameters Correlated with the Initial Response

We used receiver operating characteristic (ROC) analysis to determine the optimal cut-off value for the hematological and dosimetric variables associated with treatment response. We validated these values using logistic regression. The test results are summarized in Table 2. Baseline Hb level, lymphocyte-related hematological parameters, heart dosimetric parameters, mean lung dose, and cumulative cisplatin dose were significantly associated with treatment response in univariate regression analysis. In multivariate regression, baseline Hb > 11.25 and ≤11.25 (OR 11.536, 95% CI 2.036–65.377, *p* = 0.006), the highest NLR during CCRT > 14.48 and ≤14.48 (OR 0.261, 95% CI 0.075–0.910, *p* = 0.035), heart V10 > 86.5% and ≤86.5% (OR 0.278, 95% CI 0.079–0.976, *p* = 0.046), and cumulative cisplatin dose >147.5 mg/m^2^ and ≤147.5 mg/m^2^ (OR 4.966, 95% CI 1.446–17.051, *p* = 0.011) remained significantly associated with the initial response.

## 4. Discussion

In this retrospective cohort study, we demonstrated the value of initial clinical response, overall survival, and possible variables correlated with response in patients with locally advanced esophageal cancer receiving definitive CCRT. The factors predicting survival and treatment response were: (i) the highest NLR during CCRT, and (ii) cardiac dose. Despite the heart V10 failing to validate the significance in this cohort’s multivariate analysis of OS, it has been proposed as an independent prognostic factor [13,15,18]. 

The pathological distinction between squamous cell carcinoma and adenocarcinoma inherently diverges in the epidemiology, risk, and prognosis of esophageal cancer. Several categories of parameters that could predict survival have been evaluated and explored, including patient characteristics, genetic alterations, and treatment-specific variations [23,24,25,26]. Traditionally, we focused on the classical parameter defined in the AJCC staging grouping, like primary tumor invasiveness, the number of lymph node metastases, and whether it was distant disease [27]. However, if patients are not amenable to undergoing surgery, the prognostic value may be lower, owing to the nature of clinical evaluation rather than pathological analysis [28,29]. It was also proposed that other parameters, such as tumor subsite, histology, grade, NLR, age, performance status, comorbidities, smoking or drinking history, or socioeconomic status, would impact survival [26,30,31,32,33,34]. In this analysis, we confirmed the prognostic value of sex, performance status, NLR-h, and treatment response in non-operative patients. The initial clinical response is paramount, although it is limited by the accuracy of clinical evaluation [28,35]. The SANO (Surgery As Needed for Oesophageal Cancer (SANO) study group proposed and validated a comprehensive clinical response evaluation with endoscopic ultrasonography, EGD with bite-on-bite biopsies, fine-needle aspiration of suspicious lymph nodes, and PET/CT to evaluate the active surveillance strategy [36]. We used chest CT scans and the RECIST criteria as the most relevant response evaluation methods in patients with advanced disease receiving treatment. Although the response may need to be reevaluated in the era of immunotherapy, it remains a crucial intermediate endpoint to guide treatment adjustment with either consolation chemotherapy or delayed surgery [37,38,39]. The difference in median overall survival stratified by responders and non-responders in our cohort was 12 months, with a dismal prognosis among non-responders. Efforts should be made to avoid incorrect treatment.

Among the parameters associated with clinical treatment response, we found that baseline Hb, cumulative cisplatin dose before CCRT completion, NLR-h, and heart V10 were significantly associated. The baseline and cumulative cisplatin dose, modified by aggressive supportive management to enhance treatment compliance, may further impact survival outcomes [40,41]. The NLR is a robust biomarker for survival outcome prediction [42]. Neutrophils, specifically tumor-associated neutrophils (TANs), play an essential role in both direct tumoricidal and tumor microenvironments. The interaction between the antitumoral (N1 phenotype) and protumoral (N2 phenotype) phenotypes remains elusive and requires further study, especially in relation to other treatment modalities, such as RT and immunotherapy [43]. Lymphocytes, the most vulnerable white blood cells when exposed to radiation [44], are also crucial for antitumor activity and the tumor microenvironment. It is difficult to explain the whole picture of the pro- or anti-tumorigenic host response to cancer therapies with a simple NLR-h level [45]; however, it may still reflect dynamic immune alterations during treatment and predict further prognosis. RT has been correlated with negative effects on immunity in non-small cell lung cancer [46], and we found that the volume of low-dose thoracic irradiation affects systemic inflammation-immunity status, as the absolute lymphocyte count is the most correlated with decreasing blood cell count [47]. It has been proposed that radiation-induced lymphopenia impacts complete pathological response rates, and that the cardiac dose is a leading dosimetric factor associated with severe lymphopenia [17]. As the heart is the center of the circulatory system, cardiac irradiation may contribute most to lymphopenia. Therefore, it may confer a poor treatment response and ultimately result in poor survival outcomes. This study is the first to show a correlation between the treatment response to NLR-h and the cardiac dose. However, further prospective cohort studies are required to confirm these findings. Also, in the era of immunotherapy, the use of immune checkpoint inhibitors has already been integrated into esophageal cancer treatment [4,48]; whether we can further extrapolate the use of NLR to different endpoints, such as immunotherapy-related toxicity [49], requires further evaluation.

The volume of irradiated normal organs was validated with the influence of survival and was the most pronounced from the prospective evidence in RTOG 0617 for thoracic radiation. This showed a potentially detrimental effect of excessive thoracic irradiation, as Heart V5 and V30 were found to be prognostic factors for survival [50,51]. Further technical analyses also showed the dosimetric advantage of IMRT despite the larger treatment volume and advanced disease. IMRT reduced heart V40, which was significantly associated with OS in the adjusted analysis [52]. Recent studies have also demonstrated the importance of the heart subregion dose, as the dose to the base of the heart is the most relevant factor [53]. In esophageal cancer, the cardiac dose is not only correlated with severe cardiac events but also with survival outcomes [13,15,18,19,54]. We did not observe excessive cardiac events in our cohort, which may be affected by the nature of locally advanced disease, and disease progression ultimately drives patient survival. Any level of cardiac dose, as well as the mean dose, showed a strong relationship with treatment response. We believe that in a patient treated with non-surgical management, the heart dose was of greater importance than the lung dose in terms of treatment response and survival. With advances in RT techniques, proton therapy has provided benefits in sparing radiation to adjacent structures with the physical characteristics of the Bragg peak [55]. It has shown treatment results comparable with those of photon therapy, with a prominent effect in reducing the severity of treatment toxicity [56,57]. Among these toxicities, lymphopenia has drawn attention in the era of immunotherapy and is also prognostic in survival outcomes [17,58]. Proton therapy can play a role in mitigating lymphopenia when compared with photon therapy [59,60,61], and we hope the prospective NRG-GI006 (NCT03801876) and PROTECT (NCT05055648) clinical trials could validate our hypothesis. Cardioprotective agents, whether pharmacological or natural compounds, could reduce cardiovascular effects and warrant further research in this context [62]. The current cardio-oncology consensus suggests risk-stratified management with relevant prevention, treatment, and surveillance [63].

This single-institute, retrospective study has some limitations. First, the clinical treatment response was subjective, and although efforts have been made, a blinded central review may be more standardized. Second, the volumes of the irradiated thoracic organs were strongly correlated with each other, which may have confounded the results. Although in logistic regression of heart dosimetry with treatment response, heart V30 and V40 were also statistically significant, we chose V10 for further multivariate analysis because we found that a low-dose irradiated volume was more prognostic for survival. Third, we did not subscale heart structure and only analyzed the whole heart as an at-risk organ. However, evaluation is much easier and may be closer to daily practice. Finally, the timing of CBC checkups during treatment was not strictly defined. However, we performed regular follow-ups with our patients and arranged laboratory tests as indicated.

We only analyzed the laboratory data collected at baseline and during treatment; further research should focus on follow-up changes to evaluate the repair and recovery process with treatment response, although lymphocyte recovery may not alter clinical outcomes [64].

## 5. Conclusions

In conclusion, our study demonstrated that the clinical treatment response is a critical intermediate endpoint in the treatment and prediction of survival outcomes. Every effort should be made to enhance treatment response, not only using aggressive supportive management to facilitate treatment adherence but also by paying attention to the possibility of treatment-related lymphopenia. RT may be a double-edged sword; however, technological advances have made it possible to determine the underlying mechanisms and potential solutions. If non-operative management is selected, the cardiac dose should be considered, as we are moving toward an era of organ preservation.

## Figures and Tables

**Figure 1 cancers-15-04580-f001:**
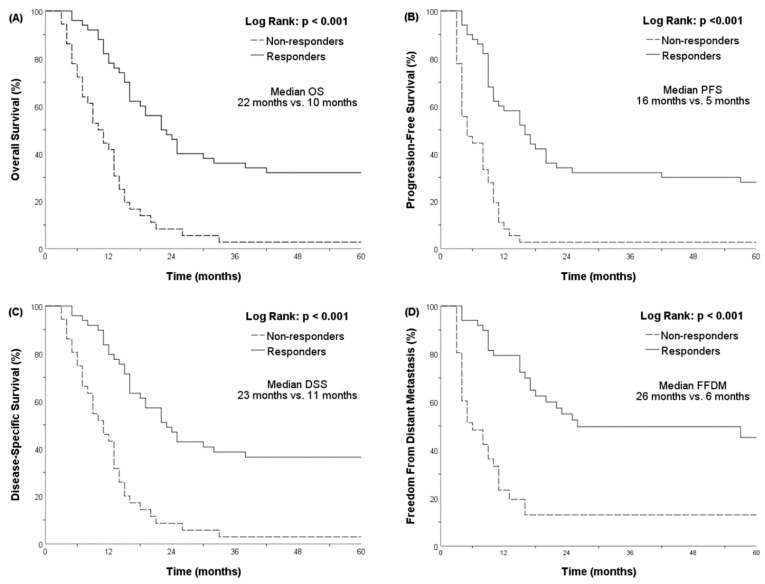
Kaplan–Meier’s curves for different survival outcomes, and patients are stratified by treatment responders (solid line) or treatment non-responders (dotted line): (**A**) overall survival (OS); (**B**) progression-free survival (PFS); (**C**) disease-specific survival (DSS); (**D**) freedom from distant metastasis (FFDM).

**Table 1 cancers-15-04580-t001:** Patient and treatment characteristics; their univariate and multivariate correlations (HR [95% CI] and *p*-value) with overall survival.

Variables (*n* = 86)	Number or Median	Standard Deviation or %	Univariate Analysis		Multivariate Analysis	
			HR [95% CI]	*p* Value	HR [95% CI]	*p* Value
Age at diagnosis (years)	60	10.4	0.992 [0.971–1.013]	0.444		
Gender						
Female	10	11.6	1		1	
Male	76	88.4	2.497 [1.077–5.791]	0.033	9.105 [2.379–34.853]	0.001
Smoker						
No (never smoked)	14	16.3	1		1	
Yes (current smoker or quitted)	60	69.8	2.020 [1.026–3.979]	0.042	0.422 [0.155–1.151]	0.092
Unknown	12	13.9				
Alcohol drinker						
No (never or not regular)	8	9.3	1		1	
Yes (current use or ever regularly use)	67	77.9	3.102 [1.122–8.578]	0.029	1.982 [0.506–7.771]	0.326
Unknown	11	12.8				
Betel-nuts chewer						
No (never or not regular)	26	30.2	1			
Yes (current use or ever regularly use)	45	52.3	1.161 [0.684–1.969]	0.580		
Unknown	15	17.5				
Eastern Cooperative Oncology Group—Performance Status, *n* (%)						
0 and 1	79	91.9	1		1	
2	7	8.1	2.201 [1.000–4.846]	0.050	3.755 [1.477–9.547]	0.005
Age Adjusted Charlson’s Comorbidity Index (excluding esophageal cancer)	2	1.4				
0–2	59	68.6	1			
>2	27	31.4	0.862 [0.5331–1.399]	0.547		
Primary tumor location, *n* (%)						
Cervical and Upper	24	27.9	1	0.399		
Middle and Lower	62	72.1	1.300 [0.784–2.157]	0.309		
Primary tumor size (cm)	5	3.5	1.039 [0.966–1.117]	0.303		
Histologic grade, *n* (%)						
Grade 1	2	2.3	1	0.901		
Grade 2	60	69.8	1.390 [0.337–5.729]	0.648		
Grade 3	5	5.8	1.352 [0.261–6.989]	0.719		
Unknown	19	22.1				
Clinical T stage, *n* (%)						
T1 and T2	32	37.2	1			
T3 and T4	54	62.8	1.384 [0.864–2.217]	0.176		
Clinical N stage, *n* (%)						
N0 and N1	41	47.7	1			
N2 and N3	45	52.3	1.459 [0.926–2.300]	0.103		
Clinical TNM stage, *n* (%)						
II	25	29.1	1			
III	61	70.9	1.399 [0.845–2.316]	0.192		
Baseline Hb (g/dL)	12.8	1.9	0.985 [0.875–1.108]	0.799		
Baseline WBC (cells/mm^3^)	7900	2669.1	1.062 [0.973–1.159]	0.180		
Baseline ANC (cells/mm^3^)	5074	2492.1	1.001 [1.000–1.002]	0.118		
Baseline ALC (cells/mm^3^)	1635	599.8	1.000 [1.000–1.000]	0.831		
Baseline NLR	3.32	2.2	1.066 [0.966–1.176]	0.207		
ALC nadir during CCRT (cells/mm^3^)	219.4	172.3	1.000 [ 0.998–1.001]	0.471		
Highest NLR during CCRT (NLR-h)	17.1	28.3	1.007 [1.000–1.015]	0.061	1.012 [1.002–1.022]	0.016
Radiotherapy technique						
3D-CRT	20	23.2	1	0.669		
IMRT	33	38.3	0.801 [0.450–1.429]	0.453		
VMAT	33	38.3	0.781 [0.438–1.392]	0.402		
Radiation dose (cGy)	5940	605.3	0.984 [0.947–1.023]	0.422		
Mean heart dose (cGy)	2047	692.5	1.000 [1.000–1.001]	0.093 *		
Dose–volume of heart (%)						
V10	90	25.8	1.010 [0.999–1.020]	0.071 *	1.012 [0.999–1.026]	0.072
V20	44	23.1	1.007 [0.997–1.017]	0.182		
V30	15.5	12.7	1.013 [0.996–1.031]	0.124		
V40	5	5.5	1.033 [0.993–1.076]	0.108		
Mean lung dose (cGy)	1511.5	295.9	1.000 [0.999–1.001]	0.670		
Dose–volume of lung (%)						
V5	91	14.9	1.005 [0.990–1.021]	0.509		
V10	69	16.8	1.001 [0.988–1.015]	0.836		
V20	21	8	0.997 [0.967–1.028]	0.832		
Radiotherapy and chemotherapy sequence, *n* (%)						
CCRT and +/− adjuvant chemo	36	41.9	1	0.608		
Induction chemo, CCRT, and +/− adjuvant chemo	48	55.8	0.797 [0.503–1.263]	0.334		
Sequential RT and chemo	2	2.3	1.053 [0.251–4.413]	0.944		
Concurrent cisplatin dose (mg/m^2^) during CCRT	75	41.6	0.998 [0.992–1.003]	0.430		
Cumulative cisplatin dose (mg/m^2^) before CCRT finished	140	54.7	0.998 [0.993–1.003]	0.434		
Treatment Response						
Responders	50	58.1	1			
Complete response	4	4.6	1	<0.001		
Partial response	46	53.5	0.888 [0.316–2.498]	0.822		
Non-responders	36	41.9	3.201 [1.985–5.164]	<0.001	4.172 [2.296–7.583]	<0.001
Stable disease	14	16.3	1.828 [0.592–5.646]	0.294		
Progressive disease	22	25.6	5.011 [1.664–15.091]	0.004		

Abbreviations: HR, hazard ratio; CI, confidence interval; Hb, hemoglobin; WBC, white blood cell count; ANC, absolute neutrophil count; ALC, absolute lymphocyte count; NLR, neutrophil-to-lymphocyte ratio; CCRT, concurrent chemoradiation; 3DCRT, 3-dimensional conformal radiation therapy; IMRT, intensity-modulated radiation therapy; VMAT, volumetric modulated arc therapy; Vx (%), relative percent of volumes for at least x (Gy); chemo, chemotherapy. * Both mean heart dose and heart V10 showed statistical significance in univariate analysis with overall survival; however, with a strong correlation in between, we chose heart V10 for further multivariate analyses.

**Table 2 cancers-15-04580-t002:** Logistic regression of hematological and treatment variable correlation with treatment response.

Variables	Youden’s Index Cutoff Value	OR	95% CI	Univariate *p* Value	OR	95% CI	Multivariate *p* Value
Baseline Hb	11.25	5.750	1.673–19.761	0.005	11.536	2.036–65.377	0.006
Baseline NLR	4.34	0.370	0.141–0.976	0.045 *			
Highest NLR during CCRT	14.48	0.264	0.101–0.690	0.007 *	0.261	0.075–0.910	0.035
ALC nadir during CCRT	137.8	2.533	1.006–6.381	0.049 *			
Mean heart dose (cGy)	1964	0.327	0.130–0.826	0.018 **			
Heart V10	86.5	0.214	0.076–0.607	0.004 **	0.278	0.079–0.976	0.046
Heart V20	58.5	0.400	0.157–1.022	0.055			
Heart V30	15.5	0.293	0.117–0.734	0.009 **			
Heart V40	4.5	0.234	0.092–0.598	0.002 **			
Mean lung dose (cGy)	1808	0.173	0.034–0.888	0.036	0.353	0.030–4.142	0.407
Lung V5	88.5	1.737	0.722–4.178	0.218			
Lung V10	71.5	0.503	0.210–1.205	0.123			
Lung V20	20.5	0.542	0.227–1.295	0.168			
Cumulative cisplatin dose (mg/m^2^) before CCRT finished	147.5	2.857	1.118–7.303	0.028	4.966	1.446–17.051	0.011

Abbreviations: OR, odds ratio; CI, confidence interval; Hb, hemoglobin; NLR, neutrophil-to-lymphocyte ratio; CCRT, concurrent chemoradiation; ALC, absolute lymphocyte count; Vx (%), relative percent of volumes for at least x (Gy). *, ** Multiple lymphocyte-related hematological parameters and heart dosimetric parameters showed statistical significance in univariate analysis with treatment response; however, with a strong correlation within each category, we chose the highest NLR during CCRT and heart V10 to further multivariate analysis.

## Data Availability

The data presented in this study are available on request from the corresponding author, upon reasonable request.

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
