# Peer review of "Cardiac Dose Predicts the Response to Concurrent Chemoradiotherapy in Esophageal Squamous Cell Carcinoma"

_cancers, 2023, doi:10.3390/cancers15184580_

Round 1

Reviewer 1 Report

The manuscript "Cardiac Dose Predicts the Response to Concurrent Chemoradiotherapy in Esophageal Squamous Cell Carcinoma" is devoted to the evaluation of the treatment of inoperable forms of esophageal cancer. Colleagues carried out a very important and in-depth study of the clinical material. Statistical processing of the material reflects the dependencies and patterns of response to therapy, and is also valid and professionally performed. The only minor remark in the opinion of the reviewer is the insufficient explanation that the response of patients of different sexes to this therapy has differences. I would like a description of the rationale for this fact, although, of course, it may be inexplicable. Also, if the authors consider it possible, it would be good to speculate about the different responses to therapy due to the difference in the intensity of the repair processes in different patients. It is this fact that most often affects the progression of tumors in the application of radiotherapy. All the above remarks do not detract from the importance and significance of this work. The work deserves to be published in this journal.

Reviewer 2 Report

Manuscript titled “ Cardiac Dose Predicts the Response to Concurrent Chemoradiotherapy in Esophageal Squamous Cell Carcinoma     “ is a very interesting articlein the field of cardioncology.

The overall structure is of good quality and easy to read. Methods and Results are clear and results corroborate the initial hypothesis of the authors. Figures and Tables are of sufficient quality and easy to read as well as to understand to readers. However, manuscript need some improvements, specifically in Introduction and/or Discussion. Here the points:

1. Authors should add a small description on the natural and pharmacological cardioprotective drugs that could be used in cancer patients before and during radio and chemotherapy in order to reduce cardiovascular effects. For example, sacubitril valsartan, gliflozins, vericiguat and natural flavonoids ( quercetin that is very useful in patients with cancer and cardiovascular diseases: cite 10.1007/s10973-017-6135-5 )

2. Could neutrophil to lymphocyte ratio also a useful predictor of immune checkpoint-induced cardiotoxicity? please discuss on this point

After these changes, the article could be suitable for publication in this journal.

Quality of English is appropriate and easy to understand.

Reviewer 3 Report

The goal of this research is to evaluate the clinical variables and chemoradiation parameters associated with treatment response. Actually, the current proposal is interesting and well-written. Therefore, I recommend that the current study be published after minor revisions as follows:

1-   Please add a diagrammatic figure to summarize these findings

2-   Please discuss the possible mechanisms for ‘’ Cardiac irradiation may affect the treatment response’’.

Reviewer 4 Report

Very complex paper that considers different aspects. The additional data shows the data of the multivariates. In the final version of the paper, the authors could include more data from multivariates than univariates.  Also considering the title and the results obtained, the authors could expand the introduction on the esophageal squamous cell carcinoma and cardiac dose (add a background on this using the data in the literature).
